

# The yeast telomerase RNA, TLC1, participates in two distinct modes of TLC1-TLC1 association processes *in vivo*

Tet Matsuguchi and Elizabeth Blackburn

Department of Biochemistry and Biophysics, University of California, San Francisco, CA, United States

## ABSTRACT

Telomerase core enzyme minimally consists of the telomerase reverse transcriptase domain-containing protein (Est2 in budding yeast *S. cerevisiae*) and telomerase RNA, which contains the template specifying the telomeric repeat sequence synthesized. Here we report that *in vivo*, a fraction of *S. cerevisiae* telomerase RNA (TLC1) molecules form complexes containing least two molecules of TLC1, via two separable modes: one requiring a sequence in the 3′ region of the immature TLC1 precursor and the other requiring Ku and Sir4. Such physical TLC1-TLC1 association peaked in G1 phase and did not require telomere silencing, telomere tethering to the nuclear periphery, telomerase holoenzyme assembly, or detectable Est2-Est2 protein association. These data indicate that TLC1-TLC1 associations reflect processes occurring during telomerase biogenesis; we propose that TLC1-TLC1 associations and subsequent reorganization may be regulatory steps in telomerase enzymatic activation.

## INTRODUCTION

Telomeric DNA is typically composed of repetitive sequences (TG1-3 repeats in the budding yeast *S. cerevisiae*) that allow recruitment of specialized macromolecular complexes that help replenish and protect telomeres (*De Lange, Lundblad & Blackburn, 2006*). These include the ribonucleoprotein telomerase, which adds telomeric DNA by the action of its reverse transcriptase-containing subunit (Est2 in *S. cerevisiae*), templated by a sequence within the telomerase RNA component (TLC1 in *S. cerevisiae*), as well as telomere-protective double-stranded and single-stranded telomeric DNA binding proteins, such as Rap1 and Cdc13 in yeast (*Jain & Cooper, 2010*).

Budding yeast telomerase RNA, TLC1, is over 1,300 nucleotides in size and, in addition to providing the template for reverse transcription, has extensive secondary structures (*Zappulla & Cech , 2004*). Certain structures within TLC1 have been defined and form binding sites for Est2 and other telomerase factors. The critical central core of TLC1 includes a structurally highly conserved pseudoknot to which Est2 binds, while an Sm-protein binding site is located near the 3′ end, which is important for the stability and processing of immature TLC1 (*Seto et al., 2002*; *Zappulla & Cech , 2004*; *Lin et al., 2004*; *Jiang et al., 2013*) (Fig. 1A). Previously, it was reported that mutations (*tlc1-42G* and

Corresponding author
Elizabeth Blackburn,
elizabeth.blackburn@ucsf.edu

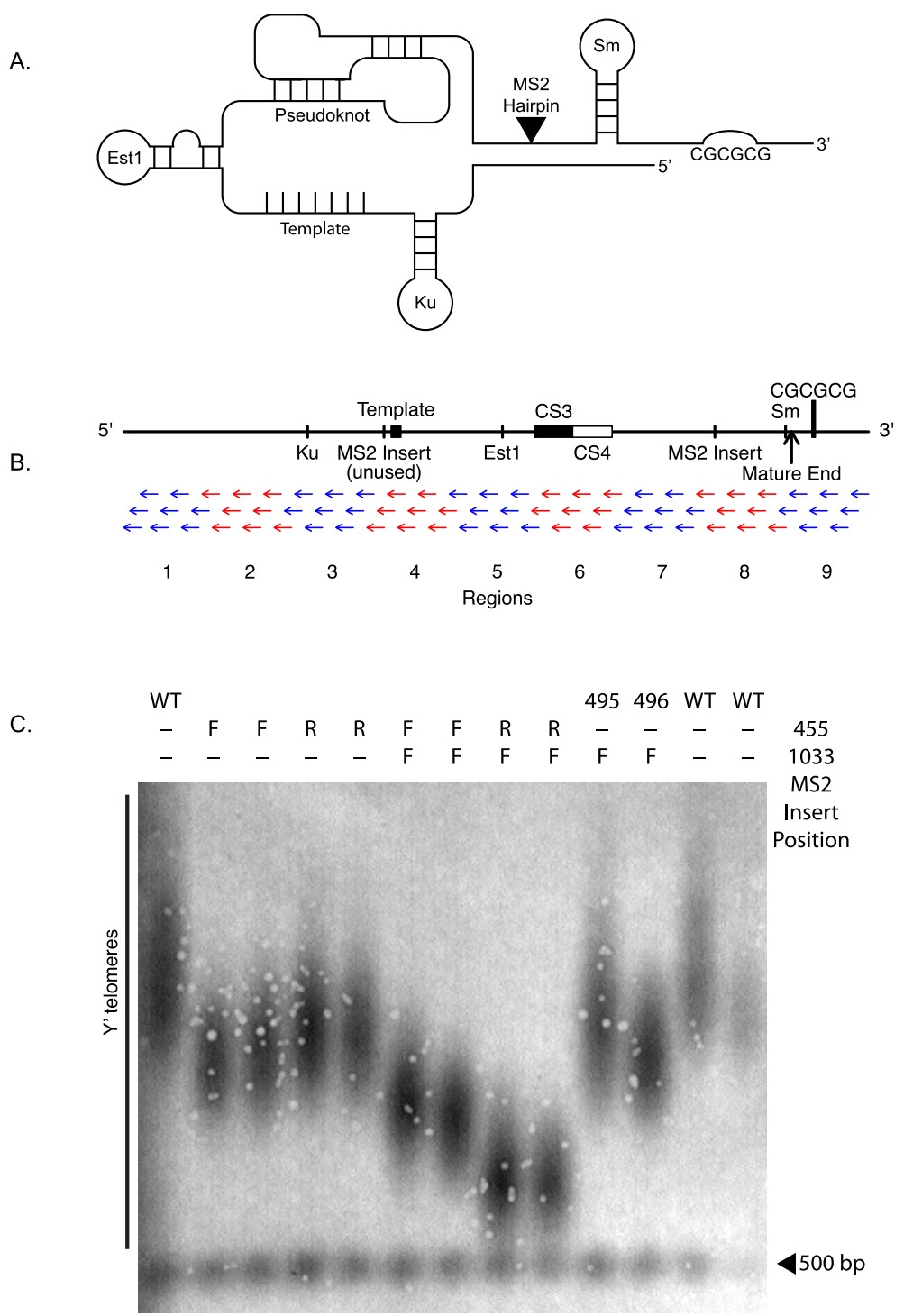

**Figure 1  TLC1-TLC1 features and MS2 hairpin insertion.** (A) Schematic and linear maps of relevant features of TLC1 RNA coding sequence before it is polyadenylated and cleaved (+1–1301). TmG: 5′ trimethylG cap (*Franke, Gehlen & Ehrenhofer-Murray, 2008*); The binding sites for Ku (+288–335), Est1 (+660–664), and Sm (+1153–1160) proteins are indicated. 

**Figure 1 (…continued)**
Telomeric template: (+468–484); CS3 (+719–784) and CS4 (+785–853): two sequences, conserved in budding yeasts, that form two sides of a stem of the evolutionarily conserved telomerase RNA pseuduoknot structure (*Lin et al., 2004*); MS2: site of the tandem inserted MS2 coat protein-binding hairpins used in this work, at BclI site (+1033); cleavage site: the 3′ end of the mature TLC1 (+1167); CGCGCG: sequence (+1204) previously implicated in TLC1 *in vitro* dimerization, located in the cleaved-off 3′ extension of pre-processed immature TLC1 RNA (*Chapon, Cech & Zaug, 1997*). (B) Anti-sense oligonucleotides targeted against the full length of TLC1. Each of the 72 anti-sense oligonucleotides are 30 bases in length and overlap with each other by 2–5 bases. The oligos are divided into 9 groups (alternating set of blue and red) of 8 oligos. (C) The MS2 RNA binding hairpins were inserted at NcoI (nucleotide position 455) or BclI (nucleotide position 1033) of TLC1 RNA coding gene. The telomere Southern blot of Y′ telomeres shows the impact of the MS2 insert at different positions. The table at the top denotes where the MS2 was inserted (at either nucleotide positions 455 or 1033). F and R denote whether the MS2 hairpin was inserted in correct (F) or in reverse (R) direction. The MS2 insertion at 1033 position seem to have the least impact on the telomere length (yEHB22,495/496). WT = yEHB22,321/465.

*tlc1-42C*) in a 6-base palindromic sequence, located within the TLC1 precursor 3′ region that is cleaved off to form the processed mature TLC1 RNA (see Fig. 1A), cause telomeres to be shorter *in vivo* and abrogate dimerization of TLC1 precursor synthesized *in vitro* (*Gipson et al., 2007*). Additionally, a 48-nucleotide stem motif in TLC1 directly binds the Ku70/Ku80 complex, which, in addition to its widely conserved canonical role in non-homologous end joining (NHEJ), is required for many aspects of yeast telomere biology (*Stellwagen et al., 2003*). This TLC1-Ku interaction, while not absolutely required for telomere maintenance by telomerase *in vivo*, is required for maintenance of full-length telomeres, *in vivo* association of Est2 to telomeres in G1 phase of the cell cycle (*Fisher, Taggart & Zakian, 2004*), full recruitment of telomeres to the nuclear periphery (*Taddei et al., 2004*), and transcriptional silencing at telomeres (*Boulton & Jackson, 1998*). A mutant Ku containing a small insertion, *yku80-135i*, specifically abrogates the TLC1-Ku interaction but leaves NHEJ intact (*Stellwagen et al., 2003*). Est1 and Est3 are essential factors for telomerase, which together with Est2 and TLC1, make up the telomerase holoenzyme. Est1 associates with the telomerase complex by directly binding to a bulge-stem region of TLC1 conserved in several budding yeasts, and this association is critical for the recruitment of telomerase to telomeres (*Seto et al., 2002*; *Chan, Boulé & Zakian, 2008*).

Human, *S. cerevisiae*, and *Tetrahymena* (ciliated protozoan) telomerases have been inferred to be active as a monomer *in vitro* (*Bryan, Goodrich & Cech, 2003*; *Alves et al., 2008*; *Shcherbakova et al., 2009*; *Jiang et al., 2013*). However, reports have also suggested that the human, *S. cerevisiae*, and *Euplotes* (ciliated protozoan) telomerase complexes can exist in a dimeric (or other oligomeric) forms (*Prescott & Blackburn, 1997*; *Wenz et al., 2001*; *Beattie et al., 2001*; *Wang, Dean & Shippen, 2002*). Recent single-molecule electron microscopic structural determinations indicate that core human telomerase complex (telomerase RNA, hTER, and reverse transcriptase, hTERT) is a dimer *in vitro* held together by RNA-RNA (hTER-hTER) interaction (*Sauerwald et al., 2013*).

Here, we explored possible modes of physical telomerase dimerization *in vivo*, focusing on the yeast telomerase RNA component TLC1. We developed a biochemical method that directly demonstrates a physical TLC1-TLC1 association (dimerization/oligomerization; direct or indirect), quantified in extracts of cells expressing normal amounts of telomerase

RNA from the endogenous *TLC1* gene chromosomal locus. We have not determined whether there are more than two molecules of TLC1 that are associated in complexes, so for simplicity, we refer to this as TLC1-TLC1 association. We report here that such TLC1-TLC1 associations occur *in vivo* via two modes, each mode having distinctive requirements. Our evidence supports association between telomerase RNAs occurring during the biogenesis of active telomerase complex, with potential functional importance in the regulation of telomerase activity.

## MATERIALS AND METHODS

### Plasmids

The integrating vector, pRS306-TLC1, was provided by Jue Lin. The MS2 CP-binding RNA hairpins were constructed by annealing overlapping oligonucleotide in a standard PCR protocol. The hairpin construct was cloned into the BclI site of pRS306-TLC1. The fusion PCR method was used to construct *tlc1-42G* and *tlc1-42C* alleles, which were cloned between the BclI and XhoI sites of pRS306-TLC1. CEN-ARS versions of the plasmids were made by subcloning BamHI-XhoI fragments of the integrating vectors into the vector pRS316.

### Yeast strains and growth media

Yeast strains were in the S288c background and are isogenic with BY4746, except as noted in Table 1 (*Baker Brachmann et al., 1998*; *Tomlin et al., 2001*). Yeast cultures were grown in standard rich medium or minimal media (YEPD or CSM). Deletion strains were made using a PCR-based transformation method (*Longtine et al., 1998*). Cell cycle synchronization was achieved by addition of 5 uM alpha-factor for 4 h, which arrests yeast cells in G1 phase. The cells were released from the arrest by washing away media containing alpha-factor and addition of new media.

### Immunoprecipitation of MS2 hairpin-tagged TLC1

TLC1 was tagged with two MS2 coat-protein-binding RNA hairpins at the BclI restriction site in the *TLC1* coding region sequence. This gene construct with its native promoter was integrated at the endogenous chromosomal *TLC1* locus, in tandem with untagged, wild-type *TLC1*, flanking the *URA3* marker. MS2 coat protein fused to 3 Myc epitope tags was expressed either in *tlc1Δ* or in experimental strains containing both tagged and untagged *TLC1*. The MS2 coat protein from tlc1Δ strains were used for coIP experiments using strains yEHB22,807-824. Whole cell lysates were prepared from cultures in log-phase of growth in YEPD ($OD_{600} = 0.6–1.0$) using glass beads and bead beaters. The lysis/wash buffer contained 50 mM HEPES-KCl pH8.0, 2 mM EDTA, 2 mM EGTA, 0.1% Nonidet P40, 10% glycerol, cOmplete EDTA-free protease inhibitors (Roche, Basel, Switzerland) and RNasin (1 uL/mL; Promega, Madison, WI, USA). The lysate concentrations were adjusted to $A_{260nm} = 40$ before immunoprecipitation. For lysates containing co-expressed MS2 coat protein, 400 uL of lysate was mixed with 1.5 mg Dynal ProA magnetic beads (Invitrogen, Waltham, MA, USA) and 1 ug of monoclonal anti-Myc antibody (9E11; Santa Cruz Biotechnology, Santa Cruz, CA, USA). For experiments in which MS2 coat

**Table 1 Strains used.** All strains are in the S288c strain background and are isogenic to BY4746 (*Tomlin et al., 2001*), except as noted below. BAR1 gene was deleted from BY4720 strain and mated to BY4741 (*Baker Brachmann et al., 1998*). The resulting diploid strain became the parental strain for two independent spores, yEHB22,321/465 and yEHB22,322/466. The strains yEHB22,803/804 were generous gift of Jonathan Weissman. All haploid strains have independently isolated isogenic duplicates from parental strains as noted in the table below (denoted by yEHB22,XXX/YYY, which refers to two stains yEHB22,XXX and yEHB22,YYY). The diploid strains, yEHB22,825-827 were created by mating yEHB321/465 and yEHB22,322/466 strains that have Est2 fused to indicated tags. Strains that were further modified in this study are in bold.

| Strain number | Relevant genotype |
| --- | --- |
| **yEHB22,321/465** | ***ADE2 his3△1 leu2△0 lys2△0 met15△0 trp1△63 ura3△0 bar1△0 MATa*** |
| **yEHB22,322/466** | ***ADE2 his3△1 leu2△0 lys2△0 met15△0 trp1△63 ura3△0 bar1△0 MATα*** |
| yEHB22,495/496 | yEHB22,321/465 but *TLC1-MS2* |
| **yEHB22,720/721** | **yEHB22,321/465 but *HIS3-P$_{CYC1}$-CP-3xMyc*** |
| yEHB22,722/723 | yEHB22,720/721 but *TLC1-MS2* |
| **yEHB22,662/663** | **yEHB22,720/721 but *TLC1-URA3-TLC1-MS2*** |
| **yEHB22,750/751** | **yEHB22,720/721 but *TLC1-LEU2-TLC1-MS2*** |
| yEHB22,742/743 | yEHB22,720/721 but *tlc1-42G-URA3-TLC1-MS2* |
| yEHB22,744/745 | yEHB22,720/721 but *tlc1-42C-URA3-TLC1-MS2* |
| yEHB22,776/777 | yEHB22,720/721 but *tlc1-42C-URA3-tlc1-42G-MS2* |
| yEHB22,704/705 | yEHB22,662/663 but *tgs1△::KanMX6* |
| yEHB22,768/769 | yEHB22,750/751 but *nup133△::KanMX6* |
| yEHB22,698/699 | yEHB22,662/663 but *est1△::KanMX6* |
| yEHB22,724/725 | yEHB22,662/663 but *est2△::KanMX6* |
| yEHB22,700/701 | yEHB22,662/663 but *est3△::KanMX6* |
| yEHB22,682/683 | yEHB22,662/663 but *yku70△::KanMX6* |
| yEHB22,686/687 | yEHB22,662/663 but *yku80△::KanMX6* |
| yEHB22,758/759 | yEHB22,750/751 but *yku80-135i* |
| yEHB22,702/703 | yEHB22,662/663 but *arf1△::KanMX6* |
| yEHB22,706/707 | yEHB22,662/663 but *cdc73△::KanMX6* |
| yEHB22,726/727 | yEHB22,662/663 but *ctr9△::KanMX6* |
| yEHB22,764/765 | yEHB22,750/751 but *ctf18△::KanMX6* |
| yEHB22,766/767 | yEHB22,750/751 but *esc1△::KanMX6* |
| yEHB22,728/729 | yEHB22,662/663 but *sir2△::KanMX6* |
| yEHB22,762/763 | yEHB22,750/751 but *sir3△::KanMX6* |
| yEHB22,730/731 | yEHB22,662/663 but *sir4△::KanMX6* |
| yEHB22,787/788 | yEHB22,662/663 but *sir4-42::KanMX6* |
| yEHB22,770/771 | yEHB22,750/751 but *tel1△::KanMX6* |
| yEHB22,774/775 | yEHB22,662/663 but *sir4△::KanMX6 yku80△::TRP1* |
| yEHB22,776/777 | yEHB22,720/721 but *tlc1-42G-URA3-TLC1-MS2 yku80△::TRP1* |
| **yEHB22,803/804** | ***LYS2 can1△::STE2$_P$-HIS5 lyp1△::STE3$_P$-LEU2 MATα*** |
| yEHB22,805/806 | yEHB22,803/804 but *TLC1-MS2* |
| **yEHB22,807/808** | **yEHB22,803/804 but *TLC1-URA3-TLC1-MS2*** |
| yEHB22,809/810 | yEHB22,803/804 but *tlc1-42G-URA3-TLC1-MS2* |
| yEHB22,811/812 | yEHB22,803/804 but *tlc1-42C-URA3-TLC1-MS2* |
| yEHB22,813/814 | yEHB22,803/804 but *tlc1-42C-URA3-tlc1-42G-MS2* |
| yEHB22,815/816 | yEHB22,807/808 but *yku80-135i* |
| yEHB22,817/818 | yEHB22,807/808 but *sir4△::KanMX6* |
| yEHB22,819/820 | yEHB22,807/808 but *sir2△::KanMX6* |
| yEHB22,821/822 | yEHB22,807/808 but *sir4△::KanMX6 yku80-135i* |
| yEHB22,823/824 | yEHB22,803/804 but *tlc1-42G-URA3-TLC1-MS2 sir4△::KanMX6* |
| yEHB22,825 | *EST2-3xFLAG/EST2-13xMyc MATa/α* |
| yEHB22,826 | *EST2-3xFLAG/EST2 MATa/α* |
| yEHB22,827 | *EST2-3xFLAG-13xMyc/EST2 MATa/α* |
protein was purified separately, ProA magnetic beads, anti-Myc antibody, and whole cell lysate containing MS2 coat protein (at $A_{260nm} = 60$–$80$) were incubated for 1–2 h. The immunoprecipitation was allowed to take place at 4 °C for 4-hours to overnight. The immunoprecipitates were washed 3 times with the wash buffer. For oligonucleotide-directed displacement experiments, the immunoprecipitates on the beads were aliquotted in separate tubes after the first wash. Each sample was subjected to different sets of oligonucleotides diluted to 0.5 uM each in the wash buffer and incubated for 1 h. Further washes were carried out as above before RNA extraction.

## Immunoprecipitation of tagged proteins

For immunoprecipitation of tagged proteins (Est2-13xMyc, Est2-3xFLAG), lysates were prepared as described above. For Myc-tagged proteins, the lysate was mixed with 1.5 mg Dynal ProA magnetic beads, and 1 ug 9E11 antibody. For FLAG-tagged proteins, lysate was incubated with 50 uL of M2-conjugated agarose beads. For sequential immunoprecipitation of FLAG-tagged proteins followed by Myc-tagged proteins, 15 ug of 3xFLAG peptide was added to the M2-conjugated agarose beads. The eluate was then used for Myc-tag immunoprecipitation as described.

## Quantitative reverse transcription and PCR (qRT-PCR)

RNA from input and immunoprecipitates were isolated using RNeasy Mini Kit (Qiagen, Hilden, Germany), including the DNase step as described by the manufacturer. The primer set for PGK1 was designed using IDT's PrimerQuest program. The reverse primers used to distinguish tagged and untagged TLC1 were designed within and at the insertion junction, respectively, of the MS2 hairpin tag. One-step reverse transcription and PCR kits were used for all RNA quantifications, except for the quantification of immature TLC1 (Stratagene, Invitrogen, Waltham, MA, USA). For quantification of immature TLC1, or 3′ regions of TLC1, SuperScript III and random hexamer were used for reverse transcription. Subsequently, SYBR Green I Master mix kit (Roche, Basel, Switzerland) was used for quantitative PCR. All quantitative PCR runs included serially diluted RNA samples to make standard curve, from which relative quantitative values were derived using the LightCycler software. The oligonucleotide sequences used in qRT-PCR reactions are listed in Table 2.

## Calculation of fraction TLC1 in dimer form

Four quantitative values from qRT-PCR assays are used to estimate the fraction of TLC1 in dimer form: untagged and MS2-tagged TLC1 in the input lysate ($TLC1_{IN}$, $MS2_{IN}$), untagged and MS2-tagged TLC1 in MS2-immunoprecipitate ($TLC1_{IP}$, $MS2_{IP}$). The following equations are used:

$$f(TLC1) = TLC1_{IN}/(TLC1_{IN} + MS2_{IN}) \tag{1}$$

$$f(IP) = MS2_{IP}/MS2_{IN} \tag{2}$$

$$n(heterodimer) = TLC1_{IP}/f(IP) \tag{3}$$

$$f(heterodimer) = 2 * f(TLC1) * (1 - f(TLC1)) \tag{4}$$

$$n(dimer) = n(heterodimer)/f(heterodimer) \tag{5}$$

**Table 2  Primer sequences for qRT-PCR.**

| Amplicon | Primer number | Sequence (5′ to 3′) |
|---|---|---|
| *PGK1* | oEHB22,0716 | GGCTGGTGCTGAAATCGTTCCAAA |
| | oEHB22,0717[a] | AGCCAGCTGGAATACCTTCCTTGT |
| Untagged *TLC1* | oEHB22,0561 | CATCGAACGATGTGACAGAGAA |
| | oEHB22,0801[a] | GACAAAAATACCGTATTGATCATTAA |
| MS2-tagged *TLC1* | oEHB22,0563 | ATGCCTGCAGGTCGACTCTAGAAA |
| | oEHB22,0338[a] | TGCGACAAAAATACCGTATTGATCA |
| Uncleaved, untagged *TLC1* | oEHB22,1015 | TATCTATTAAAACTACTTTGATGATCAGTA |
| | oEHB22,1038[a] | AGCGATATACAAGTACAGTACGCGCG |
| Uncleaved, MS2-tagged *TLC1* | oEHB22,0339 | AGCTTGCATGCCTGCAGGTCGACTC |
| | oEHB22,1038[a] | AGCGATATACAAGTACAGTACGCGCG |
| *CLN2* | oEHB22,712 | TTGTTCGAGCTGTCTGTGGTCACT |
| | oEHB22,713[a] | AATTTGGCTTGGTCCCGTAACACG |
| *CLN3* | oEHB22,837 | AAGGCCGCTGTACAACCTGACTAA |
| | oEHB22,838[a] | TGAACCGCGAGGAATACTTGTCCA |

**Notes.**
[a] Primer used in the reverse transcription step.

$$n(\text{TLC1 in dimer}) = 2 * n(\text{dimer}) \tag{6}$$

$$f(\text{dimer}) = n(\text{TLC1 in dimer})/(\text{TLC1}_{\text{IN}} + \text{MS2}_{\text{IN}}) \tag{7}$$

In the equations above f() represents "fraction of" and n() represents "amount of." (1) Fraction of TLC1 RNAs that are untagged is calculated as the fraction untagged divided by the sum of untagged and MS2-tagged RNAs. (2) Fraction of MS2-tagged TLC1 immunoprecipitated is calculated by dividing the amount of MS2-tagged TLC1 in the precipitate by the amount of MS2-tagged TLC1 in the input lysate. (3) The amount of untagged TLC1 in the precipitate represents untagged TLC1 in the heterodimeric form with the MS2-tagged TLC1. The total amount of heterodimeric TLC1 is estimated by dividing the amount of untagged TLC1 in the precipitate by the fraction of MS2-tagged TLC1 precipitated (Eq. (2)). (4) The fraction of TLC1 dimers that are in homodimeric (untagged + untagged or MS2-tagged + MS2-tagged) and heterodimeric (untagged + MS2-tagged) are assumed to result from random associations ($f(\text{TLC1})^2$, $f(\text{MS2})^2$, $2 * f(\text{TLC1}) * f(\text{MS2})$). Therefore, the fraction of TLC1 dimers in the heterodimeric form is calculated as $2 * f(\text{TLC1}) * f(\text{MS2})$ or $2 * f(\text{TLC1}) * (1 - f(\text{TLC1}))$. (5) The total amount of dimer is calculated by dividing the number of heterodimers (Eq. (3)) by the fraction of dimers that are heterodimeric (Eq. (4)). (6) The total amount of TLC1 in calculated by doubling the amount of dimer (Eq. (5)). (7) Fraction of TLC1 in dimer form is calculated by dividing the amount of TLC1 in dimer form (Eq. (6)) by the total amount of TLC1 and MS2 in the input lysate.

## Telomere length analysis

Genomic DNA was digested with XhoI and separated on a 0.85% agarose gel. DNA was denatured and transferred to a Nylon membrane, and UV-crosslinked with a Stratalinker. The membrane was blotted with telomeric oligonucleotide (5′-CACACCCACACCACACCCACAC-3′) labeled with WellRED D3 fluorescent dye at

the 5′ end. The blotted membrane was scanned and analyzed using the Odyssey Infrared Imaging System (LI-COR). A linear plasmid containing an *S. cerevisiae* telomeric DNA sequence was included as a marker of size ∼500 bp.

# RESULTS

## Co-immunoprecipitation assays demonstrate TLC1-TLC1 association *in vivo*

To quantify the association between different TLC1 molecules in yeast whole-cell extracts, a co-immunoprecipitation (coIP) assay was developed. First, we created a tagged TLC1 RNA for immunoprecipitation using a tandem pair of RNA hairpins that specifically bind to the bacteriophage MS2 Coat Protein. This two-hairpin construct was inserted at one of two sites in regions of TLC1 previously shown to accommodate insertions of modular protein binding domains with minimal if any effect on *in vivo* functions (*Bernardi & Spahr, 1972*; *Zappulla & Cech, 2004*) (Fig. 1A). Secondly, we fused three copies of myc tag to MS2 Coat Protein and integrated this gene construct into the genome of experimental strains. Co-expression of the MS2 hairpin-tagged TLC1 (TLC1-MS2) and myc-tagged Coat Protein (CP-3myc) allowed specific immunoprecipitation of TLC1-MS2 using an anti-myc antibody. Thirdly, we developed quantitative RT-PCR assays to measure levels and recovery of TLC1, using two sets of PCR primers capable of distinguishing and specifically amplifying either the untagged TLC1 or TLC1-MS2 (Figs. 2A and 2B).

Next, we verified that the insertion of the MS2 tag did not significantly alter TLC1 functions *in vivo*. We tested the telomere lengths of strains that have MS2 hairpin insertion at two different sites, nucleotide positions 455 and 1033 of TLC1 (NcoI and BclI sites). The insertion had the least impact at nucleotide position 1033 (Fig. 1C), and we used this construct for the rest of the study. The expression level of TLC1-MS2 was comparable to untagged TLC1 (Fig. 2C). The association of TLC1-MS2 with Est2 was slightly reduced compared to untagged TLC1, consistent with telomere lengths observed in cells expressing only TLC1-MS2 (Fig. 2D).

Finally, we co-expressed TLC1-MS2 and untagged TLC1 from the endogenous TLC1 locus to test the coIP of untagged TLC1 with TLC1-MS2. As a control, an equal number of cells from two independently cultured strains expressing either only untagged TLC1 or only TLC1-MS2 were mixed prior to cell lysis ("Mix" samples in figures). We found that 50–80% of total TLC1-MS2 is immunoprecipitated from lysates made from the co-expression strain and from the mixed population. A significant enrichment of untagged TLC1 in the TLC1-MS2 immunoprecipitate was observed only in the co-expression strain and not in the mixed cell population, indicating that this assay detected bona fide *in vivo* association of separate TLC1 molecules (see 'Materials and Methods' and Fig. 2E). After adjusting for the immunoprecipitation efficiency and the fact that this coIP assay only detects heterodimer of TLC1-MS2 and untagged TLC1, we determined that in unsynchronized log phase cell populations, at least 10% of TLC1 is associated with another TLC1 *in vivo* (Fig. 2E; see 'Materials and Methods' for calculation). Interestingly, we observed that the fraction of immature TLC1 molecules present in the whole lysate (4–8%)

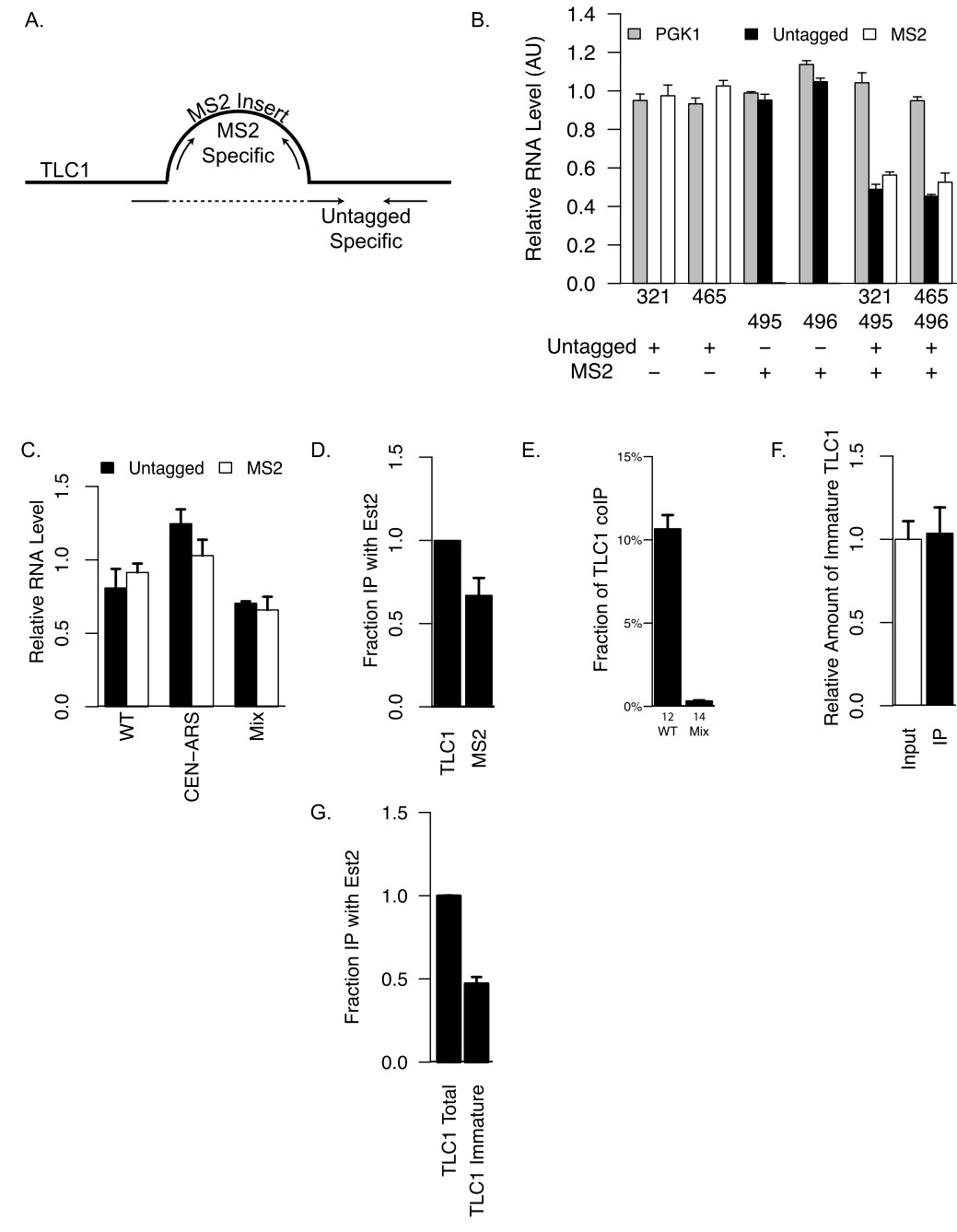

**Figure 2 Detection of TLC1-TLC1 Association.** (A) Primers used for specifically quantifying untagged or MS2-tagged TLC1 by qRT-PCR is shown schematically. The sequences of these primers are presented in Table 2. The bulge represent the MS2 sequence inserted at the BclI site of TLC1. The forward primer for specifically quantifying untagged TLC1 spans the insertion site of MS2. (B) The amounts of untagged or MS2-tagged TLC1 were quantified in strains expressing only untagged TLC1 (yEHB22,321/465) or only MS2-tagged TLC1 (yEHB22,495/496). The lysates were also mixed 1:1 in the last two groups (yEHB22,321+yEHB22,495 and yEHB22,465+yEHB22,496). PGK1 values were normalized to the average of all samples. Untagged and MS2-tagged TLC1 values were normalized to the averages of yEHB22,321/465 and yEHB22,495/496, respectively. The error bars indicate (continued on next page...)

**Figure 2 (...continued)**
the standard deviation of technical triplicates. The numbers below the *x*-axis represent the "yEHB22,xxx" strains, and the table below indicates the presence of untagged or MS2-tagged TLC1 in each sample. (C) The amounts of untagged and MS2-tagged TLC1 in total RNA normalized to PGK1 mRNA level are shown. TLC1 was expressed from the genomic locus (yEHB22,662/663) or on a CEN-ARS plasmid (pRS416-TLC1 in yEHB22,722/723). MS2-tagged TLC1 were expressed from the genomic locus. In "Mix" samples, two strains expressing tagged (yEHB22,722/723) and untagged TLC1 (yEHB22,720/721) independently were mixed before the lysis step. The error bars indicate standard deviation from two experiments. (D) The MS2 -tagged TLC1 associated with Est2. The fraction of tagged and untagged TLC1 coimmunoprecipitated with Myc tagged Est2 was quantified and normalized to the untagged TLC1 fraction. The lysates from yEHB22,321/465 strains carrying pRS426-EST2-Myc was used. The error bar indicates the standard deviation between two experiments. (E) Detection of TLC1-TLC1 association by the co-immunoprecipitation strategy. The amount of untagged TLC1 co-immunoprecipitated with MS2-tagged TLC1 was used to estimate the fraction of total TLC1 that is dimeric. The calculation method is described in the 'Methods.' Lysates from strains carrying untagged and MS2-tagged TLC1 were used (WT = yEHB22,662/663). As negative controls, separate strains carrying either MS2-tagged or untagged TLC1 (MS2-tagged = yEHB22,722/723, yEHB22,805/806, untagged = yEHB22,720/721) were mixed just before lysis. The error bars indicate the standard errors among the experiments, and the numbers just below the *x*-axis denote the number of experiments performed. (F) The immature TLC1 molecules accounts for only 4–8% of the total level of TLC1 molecules, and that this fraction was unchanged in the co-immunoprecipitated versus total TLC1. The lysates from the strains yEHB22,662/663 were used. The immature TLC1 quantity was divided by the total TLC1 amount, and the ratio was normalized to the input ratio. The error bars indicate standard deviation between two experiments. (G) The immature TLC1 associated with Est2. The fraction of TLC1, total or immature, coimmunoprecipitated with FLAG-tagged Est2 was quantified and normalized to the total TLC1 fraction. The lysates from yEHB22,826. The error bar indicates the standard deviation between two experiments.

(*Mozdy & Cech, 2006*) did not significantly change in the immunoprecipitate, indicating that both immature and mature forms of TLC1 participate comparably in TLC1-TLC1 association (Fig. 2F).

## The 3′ region of TLC1 is important for TLC1-TLC1 association

To determine the regions of TLC1 involved in the TLC1-TLC1 physical association, we designed a nucleic acid competition experiment aimed to disrupt this association by incubating the TLC1 complex(es), extracted as the immunoprecipitates from cell lysates, with excess anti-sense oligonucleotides. We designed 72 overlapping DNA oligonucleotides, each 30 bases in length, that in total were complementary to the full length of the immature TLC1, which includes the 3′ region that is cleaved off in the mature form (Fig. 1B). These oligonucleotides were incubated with the TLC1-MS2 immunoprecipitate bound to the magnetic beads in the wash buffer (see 'Materials and Methods'). We predicted that the collection of these 72 TLC1 antisense oligos would act as competitors to TLC1-TLC1 association in the immunoprecipitates. As a control, 72 different DNA oligonucleotides designed against other regions of the yeast genome were used. Incubation of the full set of 72 TLC1-antisense oligonucleotides (but not the 72 control oligonucleotides) with the immunoprecipitates reduced the amount of untagged TLC1 remaining on the affinity beads by about 70%, while not appreciably diminishing the amount of TLC1-MS2 remaining bound to the affinity beads (Figs. 3A and 3B, bottom row). This result indicated that the 72 TLC1-antisense oligonucleotides likely disrupted the association of the untagged TLC1 and TLC1-MS2.

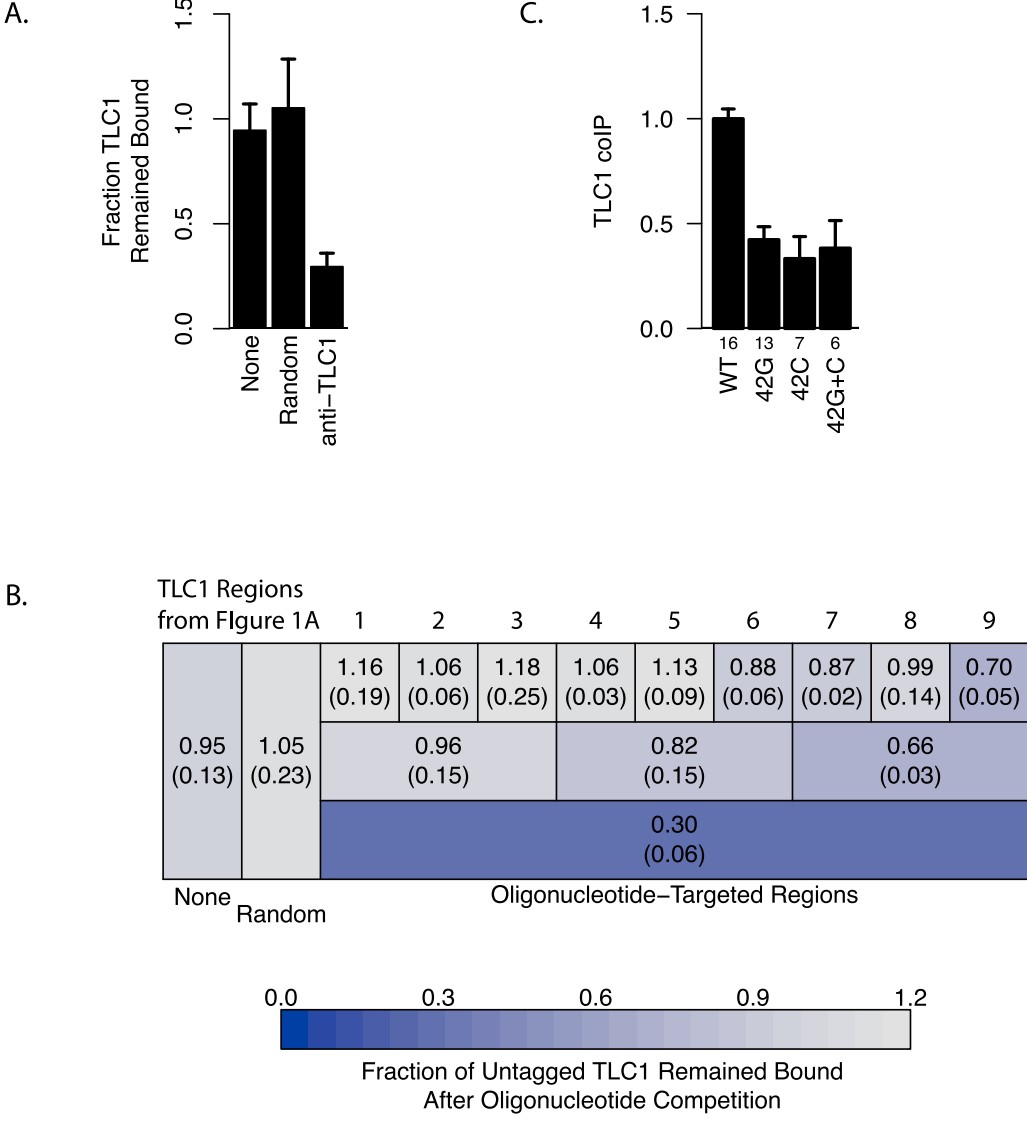

**Figure 3 Regions of TLC1 involved in TLC1-TLC1 association.** (A) Anti-sense oligonucleotides can disrupt TLC1-TLC1 association. Anti-sense oligonucleotides were designed against *TLC1* and incubated with the MS2 immunoprecipitates. The lysates from yEHB22,807/808 were used for these experiments. The fraction of untagged TLC1 that remained on the beads with the MS2-tagged TLC1 compared to the average of the sample that had no oligonucleotides or random oligonucleotides added is shown. All 72 anti-*TLC1* primers or 72 random primers were added. The error bars indicate the standard errors between the two experiments. (B) Different subsets of oligonucleotides were added during the wash step of immuno-precipitation. The lysates from yEHB22,807/808 were used for these experiments. Each box represents the TLC1 region against which oligonucleotides added targeted. Each ninth and third region contained 8 and 24 oligonucleotides, respectively. Shown in each box is the fraction of TLC1 that remained on the beads after the competition assay compared to the average of the sample that had no oligonucleotides or random oligonucleotides added (standard deviation in parentheses). (C) The TLC1 3′ region that is cleaved off plays a role in TLC1 dimerization. The fractions of TLC1 in dimer form, normalized to WT, are shown for strains that carry mutations that disrupt palindromic sequence in the 3′ region of TLC1. WT = CGCGCG (yEHB22,662/663, yEHB22,807/808), 42G = CGGGGG (yEHB22,742/743, yEHB22,809/810), 42C = CCCCCG (yEHB22,744/745, yEHB22,811/812), 42G + C = CGGGGG + CCCCCG (yEHB22,776/777, yEHB22,813/814). The error bars indicate the standard errors among the samples. The numbers of experiments represented are shown just below the *x*-axis.

To further delineate the regions important for the TLC1-TLC1 association, different subsets of oligonucleotides were used in the same experimental set-up. The 72 oligonucleotides were subdivided into intervals encompassing thirds or ninths of the length of the immature TLC1, in order to probe each TLC1 region separately (Fig. 3B). The oligonucleotides complementary to the first third (the 5′ region) of TLC1 had little effect on disrupting TLC1-TLC1 association, while the oligonucleotides against the central and 3′ region intervals had greater effects (Fig. 3B, Row 2). Even added together, the total of the effects from each of the three separate regions was significantly less than the disruptive effect seen when all 72 oligonucleotides were added simultaneously, suggesting that there is a synergistic effect in adding all oligonucleotides at once. Similarly, separately challenging the TLC1-TLC1 immunoprecipitates in this way with the anti-sense oligonucleotides encompassing each of the one-ninth regions, especially in the 5′ regions of TLC1, disrupted the TLC1-TLC1 association to even lesser extents (Fig. 3B, Row 1).

Interestingly, TLC1-TLC1 association was disrupted by 30% using the eight antisense oligonucleotides encompassing the TLC1 3′ region. Only two of these eight oligonucleotides were complementary to the last 21 bases of the mature form of TLC1; the remaining six oligonucleotides were complementary only to the 3′ extension of the un-cleaved, immature form of TLC1 (Fig. 1B). As described above, the immature TLC1 molecules accounted for only 4–8% of the total TLC1 signal in the immunoprecipitate (Fig. 2F); thus, a reduction solely of immature TLC1 precursors cannot account for the 30% disruption by the 3′ most one-ninth TLC1-complementary oligonucleotides. This result suggests that a small region (30 bases) encompassed by just two oligonucleotides had a relatively large effect in disrupting TLC1-TLC1 association of the mature form of TLC1.

Together, these findings indicated that the 3′ region of TLC1 transcript is either the most critical for TLC1-TLC1 association to occur *in vivo*, and/or the most vulnerable to subsequent *in vitro* disruption of the associated form. This *in vitro* disruption by the 3′ region-targeting oligonucleotides could have been through a direct competition of base-paired regions between two *TLC1* RNAs, through an unwinding of some structural elements of TLC1, or disruption of RNA-protein associations. Additionally, these data suggest that the TLC1-TLC1 association mostly involves tail-tail (i.e., 3′ region with 3′ region) interactions, rather than head-head (i.e., 5′ region with 5′ region) or head-tail (i.e., 5′ region with 3′ region) formations.

Prompted by the importance of the 3′ region of TLC1, we tested the potential role in TLC1-TLC1 association for a previously identified, palindromic sequence located in the 3′ region cleaved off during TLC1 maturation and thus present only in the immature, precursor TLC1 molecules. This palindromic sequence is evolutionarily conserved among budding yeast species (*Gipson et al., 2007*). Two palindrome disruption mutations (*tlc1-42G* and *tlc1-42C*) that prevent potential intermolecular base-pairing by this sequence, and the compensatory mutations (*tlc1-42G* and *tlc1-42C* in *trans*), which restore the potential for intermolecular base-pairing but not the wild-type palindromic sequence itself, have been described previously (*Gipson et al., 2007*). We found that the palindrome disruption mutations *tlc1-42G* and *tlc1-42C*, when incorporated into untagged TLC1 in the strains also expressing TLC1-MS2, reduced TLC1-TLC1 coIP by over half

(Fig. 3C). The compensatory mutations, *tlc1-42G* and *tlc1-42C* in *trans*, although restoring intermolecular base-pairing potential, failed to restore the TLC1-TLC1 coIP level (Fig. 3C). The total levels of these mutant telomerase RNAs were unchanged from wild type; hence, efficient *in vivo* association between mature TLC1 molecules requires the specific sequence—and not simply its potential for base pairing *in trans*—of a palindromic motif located in the cleaved-off 3′ portion of the TLC1 precursor. These results indicate that at least some TLC1-TLC1 association initiates during telomerase biogenesis before processing produces the mature TLC1 3′ end.

## TLC1-TLC1 association is dependent on nuclear export and is cell cycle-regulated

Maturation of telomerase RNA including 3′ end processing takes place partially in the cytoplasm (*Gallardo et al., 2008*). Interestingly, while deletion of Tgs1, which is responsible for TLC1 m3G cap formation (*Franke, Gehlen & Ehrenhofer-Murray, 2008*), had no effect on total TLC1 levels and little effect on TLC1-TLC1 association ($p > 0.05$), mutating Nup133 (required for nuclear export) (*Gallardo et al., 2008*) diminished by at least half the fraction of TLC1 in the associated form, while causing no effect on total TLC1 levels ($p < 0.05$; Fig. 4A). This finding indicated that TLC1 export into the cytoplasm may be necessary for TLC1-TLC1 association.

TLC1 maturation by 3′ end processing is reported to be active only during G1 phase of the cell cycle (*Chapon, Cech & Zaug, 1997*). To test whether TLC1-TLC1 association is controlled during the cell cycle, yeast cell lysates were prepared at 15-minute intervals from cells following release into G1 phase from an alpha-factor arrest. Cell cycle progression and synchrony were confirmed by analysis of the various cyclin mRNA levels throughout the time course (Fig. 4B). Consistent with a previous report (*Mozdy & Cech, 2006*), the total TLC1 steady-state levels showed a slight increase as the cell cycle progressed (Fig. 4C). During the first cell cycle after the release from the 2-hour alpha-factor arrest, the fraction of TLC1 in dimer form in the coIP assay remained relatively constant (Fig. 4D). After mitosis, as the cell population re-entered the next G1 phase, the fraction of TLC1-TLC1 association abruptly increased 2-fold, with markedly different kinetics compared to the slow and steady accumulation of total TLC1throughout the cell cycle progression (Fig. 4D). This finding is consistent with TLC1-TLC1 association occurring during the biogenesis of telomerase complex, a process that has been detected only in G1 phase. The lack of a higher fraction of TLC1 in the dimer form during the G1 phase immediately following the release from the 2-hour alpha-factor arrest is also consistent with TLC1-TLC1 association during a biogenesis step, since in this situation, cells have been held in G1 phase, in the presence of active biogenesis machinery, for 120 min prior to the point of release from alpha-factor. We conclude that TLC1-TLC1 association is cell-cycle-controlled and highest in G1.

## Telomerase holoenzyme formation is not required for TLC1-TLC1 association

To test whether there are any protein factors that assist in maintaining the TLC1-TLC1 association, we treated the immunoprecipitates with trypsin. We found that protease

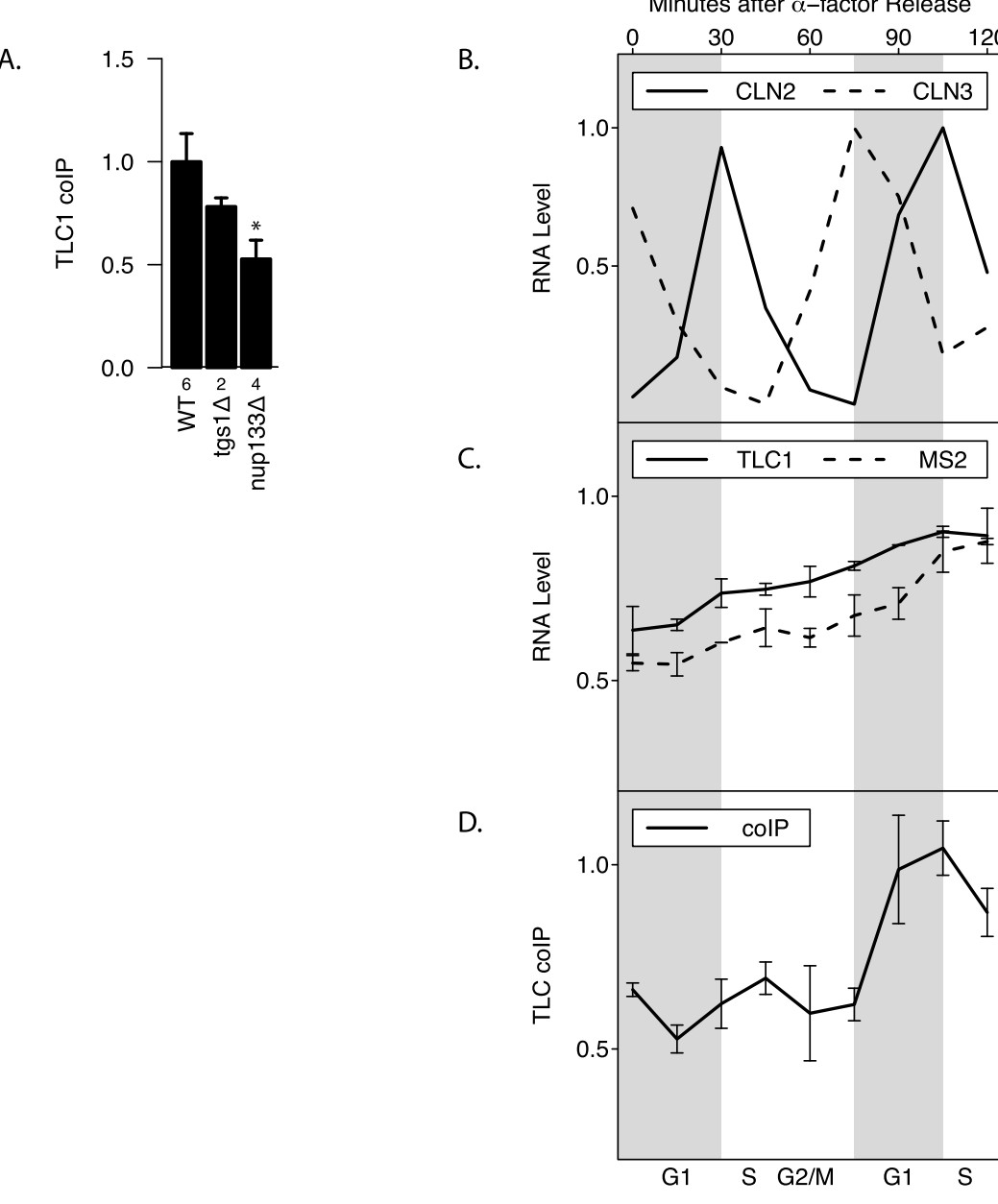

**Figure 4** **TLC1-TLC1 Association and TLC1 Biogenesis.** (A) TLC1 transport to the cytoplasm is required for TLC1-TLC1 association. The fractions of TLC1 in dimer form, normalized to WT, are shown for strains that have indicated genes involved in TLC1 biogenesis pathway is deleted. WT = yEHB22,662/663, tgsΔ = yEHB22,704/705, nup133D = yEHB22,768/769. The error bars represent standard errors among the samples, and the numbers of experiments represented are shown just below the $x$-axis. The asterisk denotes $p < 0.05$. (B) Cells (yEHB22,662/663) were arrested in alpha-factor, released and collected every 15 minutes. The first sample ($t = 0$ min) is from alpha-factor arrested cells. Levels of cyclin mRNAs measured to track cell-cycle progress. The values are normalized so that the lowest value is 0 and the highest value is 1. The horizontal bars show cell cycle phase ascertained from the measured cyclin mRNA expression levels shown. (C) Total TLC1 levels, tagged and untagged, are shown, normalized to the asynchronous culture. The error bars indicate standard deviation between two experiments. (D) The fraction of TLC1 in dimer form calculated from coIP experiments are shown. The values are normalized to the asynchronous sample, and the error bars represent the standard deviation between two experiments.

treatment reduced coIP efficiency by ∼40% compared with the control (Fig. 5A), suggesting a role for protein(s) in initiating, or stabilizing, TLC1-TLC1 association.

We tested the most likely protein factor candidate, Est2, the telomerase reverse transcriptase core protein. It has been shown that Est2 and TLC1 come together in the cytoplasm, although when in the cell cycle they initiate the interaction is unclear (*Teixeira et al., 2002*; *Gallardo et al., 2008*). In *est2Δ* strains, a diminution in TLC1-TLC1 association of about 20–25% was detected, although this measured reduction was not highly significant when compared to the control wild-type *EST2* strain ($p > 0.05$; Fig. 5B). We reasoned that the modest requirement for Est2 in TLC1-TLC1 association might be reflected in TLC1 mutants known to disrupt the core pseudoknot structure required for Est2-TLC1 interaction. Therefore, we disrupted the TLC1 pseudoknot by mutating either side of one stem (intra-base-pairing) made up of conserved sequences CS3 and CS4 (*tlc1-20* and *tlc1-21*), and restored the pseudoknot structure by the compensatory mutations (*tlc1-22*) (*Lin et al., 2004*). CoIP assays showed that the *in vivo* TLC1-TLC1 association was substantially reduced by the pseudoknot-disruptive mutations *tlc1-20* and *tlc1-21* and fully restored by the compensatory mutations, *tlc1-22* (Fig. 5C). Thus, efficient TLC1-TLC1 association requires at least this aspect of normal folding of TLC1, although binding to Est2 is largely dispensable.

Next, we tested two other essential components of the telomerase holoenzyme, Est1 and Est3, for any roles in the *in vivo* TLC1-TLC1 association. Est1-TLC1 interaction is limited to S-phase of the cell cycle, and Est3 interaction with Est2 requires Est1 and hence is also S-phase dependent (*Osterhage, Talley & Friedman, 2006*). As in the *est2Δ* strain, the *est3Δ* strain showed a modest but not significant ($p > 0.05$, Fig. 5B) reduction in TLC1-TLC1 association. In *est1Δ*, however, the TLC1-TLC1 association was reduced by ∼35% ($p < 0.05$, Fig. 5B). While many aspects of Est1 functions in telomere biology remain unclear, roles for Est1 in the recruitment of telomerase to telomeres as well as in telomerase enzymatic activation are well established (*Evans & Lundblad, 2002*). The TLC1-TLC1 association showed a somewhat greater dependence on Est1 than on Est2 and Est3. This raises the possibility that, rather than the telomerase enzymatic activation function of Est1, the telomere recruitment or other function unique to Est1 may play a role in TLC1-TLC1 association.

## Ku and Sir4, but not telomere silencing or tethering to the nuclear periphery, promote the same mode of TLC1-TLC1 association

To test whether other factors involved in telomerase recruitment to telomeres also affect TLC1-TLC1 association, we first performed the coIP assays in Ku mutant strains. In contrast to the more modest effects of the absence of essential telomerase components Est1, Est2 or Est3, 60–75% of the TLC1-TLC1 association was consistently lost in *yku70Δ* and *yku80Δ* strains, as well as in *yku80-135i* strains (Fig. 5D), which have a small insertion in Ku that specifically abrogates TLC1-Ku interaction, but leaves NHEJ intact (*Stellwagen et al., 2003*). As previously reported (*Mozdy, Podell & Cech, 2008*), in all these Ku mutant strains the steady-state level of total TLC1 was reduced by about 25–50% (Fig. 5E), and telomeres, while stable, are shorter than in wild-type. Therefore we tested two different
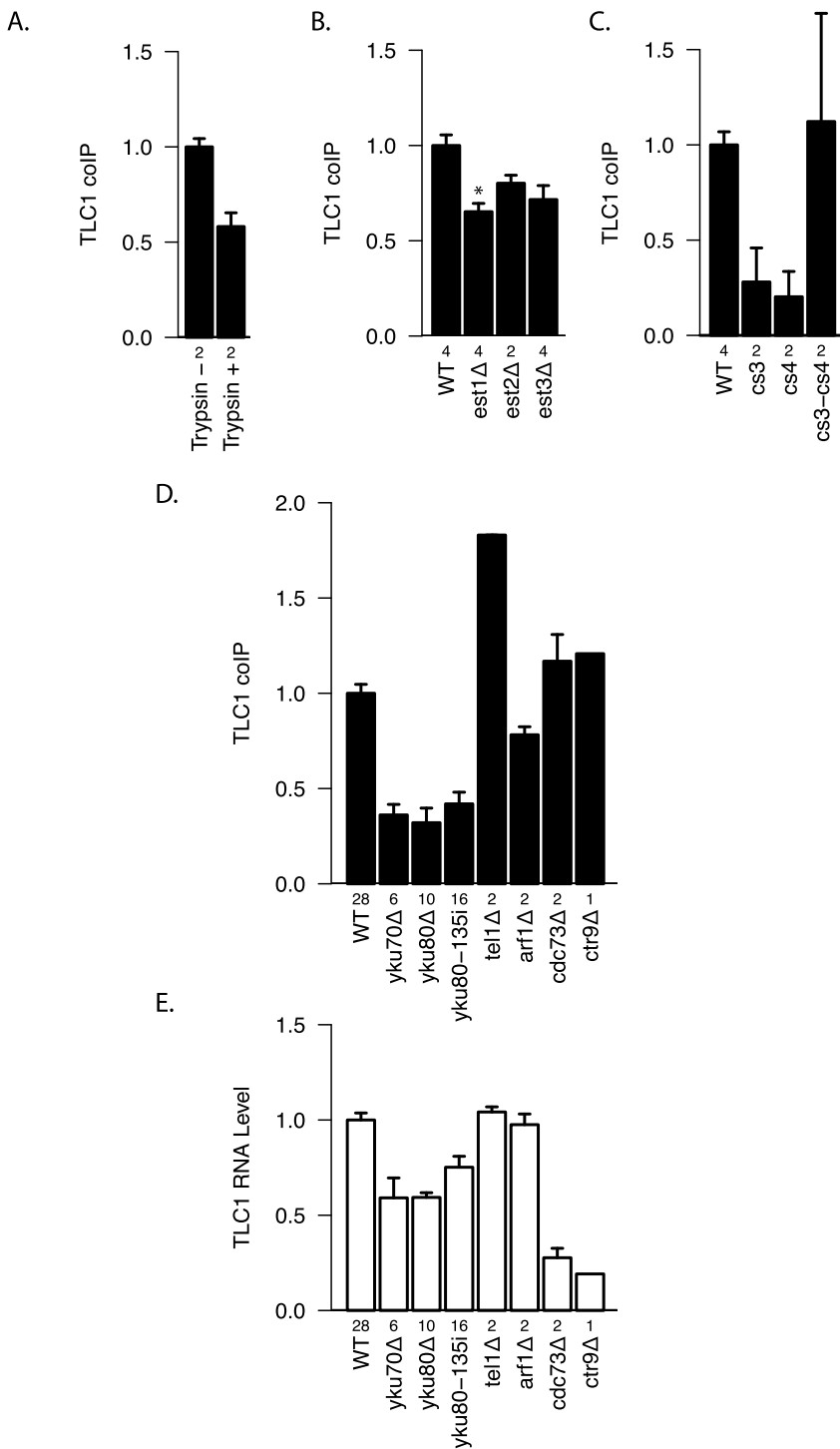

**Figure 5  Protein factor requirements for TLC1-TLC1 association.** (A) TLC1 dimerization is partially sensitive to trypsin treatment. The fraction of TLC1 that remained in the dimer form after mock or trypsin treatment was measured in lysates from WT strains (yEHB22,807/808). The values were normalized to the average of trypsin-untreated samples. The error bars indicate the standard error between two experiments. The numbers of experiments represented are shown just below the *x*-axis. (continued on next page...)

**Figure 5 (...continued)**
(B) TLC1 dimerization is only modestly affected by absence of Est 1, 2 or 3. The fractions of TLC1 in dimer form, normalized to WT, are shown for strains that have indicated gene is deleted. WT = yEHB22, 662/663, est1Δ = yEHB22,698/699, est2Δ = yEHB22,724/725, est3D = yEHB22,700/701. The error bars represent the standard deviation between samples. The numbers of experiments represented are shown just below the $x$-axis. The asterisk denotes $p < 0.05$. (C) Est2 interactions with TLC1 affect TLC1-TLC1 associations. The RNA pseudoknot structure critical for Est2 binding to TLC1 was mutated (cs3 and cs4) and compensatory mutation (cs3-cs4) was introduced. The fractions of TLC1 in dimer form, normalized to WT, are shown for strains carrying these tlc1 mutations. The strains yEHB22,722/723 carrying CEN-ARS plasmids were used: WT = pRS316-TLC1, cs3 = pRS316-tlc1-21, cs4 = pRS316-tlc1-20, cs3-cs4 = pRS316-tcl1-22. These plasmids were generous gift from Jue Lin (*Lin et al., 2004*). The error bars indicate the standard error, and the numbers of experiments represented are shown just below the $x$-axis. (D) TLC1 dimerization requires Ku. The fractions of TLC1 in dimer form, normalized to WT, are shown for strains mutated for indicated genes. WT = yEHB22,662/663, yEHB22,750/751, yEHB22, 807/808, yku70D = yEHB22,682/683, yku80Δ = yEHB22,662/663, yku80-135i = yEHB22,750/751, yEHB22,815/816, tel1Δ = yEHB22,770/771, arf1Δ = yEHB22,702/703, cdc73D = yEHB22,706/707, ctr9Δ = yEHB22,727. The error bars indicate the standard error among the samples, except for *ctr9Δ* sample, which was done only once. The numbers of experiments represented are shown just below the $x$-axis. (E) Total TLC1 levels do not determine the fraction of TLC1 in the dimer form. Untagged TLC1 levels in the total RNA were measured in strains deleted for the indicated genes. The levels were normal -ized to PGK1 mRNA levels first and then to the wild-type levels. The error bars indicate the standard error among the samples, except for *ctr9Δ* sample, which was done only once. The strains used are the same as noted in Fig. 4D. The numbers of experiments represented are shown just below the $x$-axis.

mutations (*cdc73Δ, ctr9Δ*) that reduce the steady-state level of TLC1 much more than the Ku mutations (Fig. 5E). Neither *cdc73Δ* nor *ctr9Δ* caused any decrease in the fraction of dimeric TLC1 (Fig. 5D). Furthermore, two mutations known to cause short telomeres (*arf1Δ and tel1Δ*) (*Askree et al., 2004*), also did not reduce TLC1-TLC1 association (Figs. 5D and 5E). The combined findings above indicate that Ku binding to TLC1 promotes or stabilizes TLC1-TLC1 association, and that neither reduction in TLC1 steady state level nor shorter, stable telomeres is sufficient to impair TLC1-TLC1 association.

The Ku complex is also necessary for telomere silencing (*Boulton & Jackson, 1998*) and telomere tethering to the nuclear periphery (*Taddei et al., 2004*). However, by using mutations that affect these processes, we found evidence that it is not because of these functions that Ku plays a role in TLC1-TLC1 association. Specifically, *sir3Δ* (which abrogates telomere silencing) and neither *ctf18Δ* nor *esc1Δ* (which each diminish telomere tethering) (*Hiraga, Robertson & Donaldson, 2006*) decreased TLC1-TLC1 association levels (Fig. 6A). In marked contrast, *sir4Δ* as well as *sir4-42* mutations diminished TLC1-TLC1 association to the same extent as *yku80-135i and sir2Δ to a lesser extent* (Fig. 6A). SIR4-42 mutation truncates the C-terminus of Sir4 and fails to recruit silent chromatin factors to telomeres (*Kennedy et al., 1995*). Sir4 is distinguished from the other telomere silencing Sir proteins Sir2 and Sir3 by its localization on telomeres closer to the distal tip than Sir2 and Sir3, and the Ku complex is reported to interact physically with Sir4 (*Tsukamoto, Kato & Ikeda, 1997*). Since Ku and Sir4 are localized on telomeres, we tested whether detection of TLC1-TLC1 association in cell extracts by the coIP assay was dependent on DNA. However, DNase treatment of the extracts did not diminish the fraction of TLC1 detected in dimeric form (Fig. 6B).

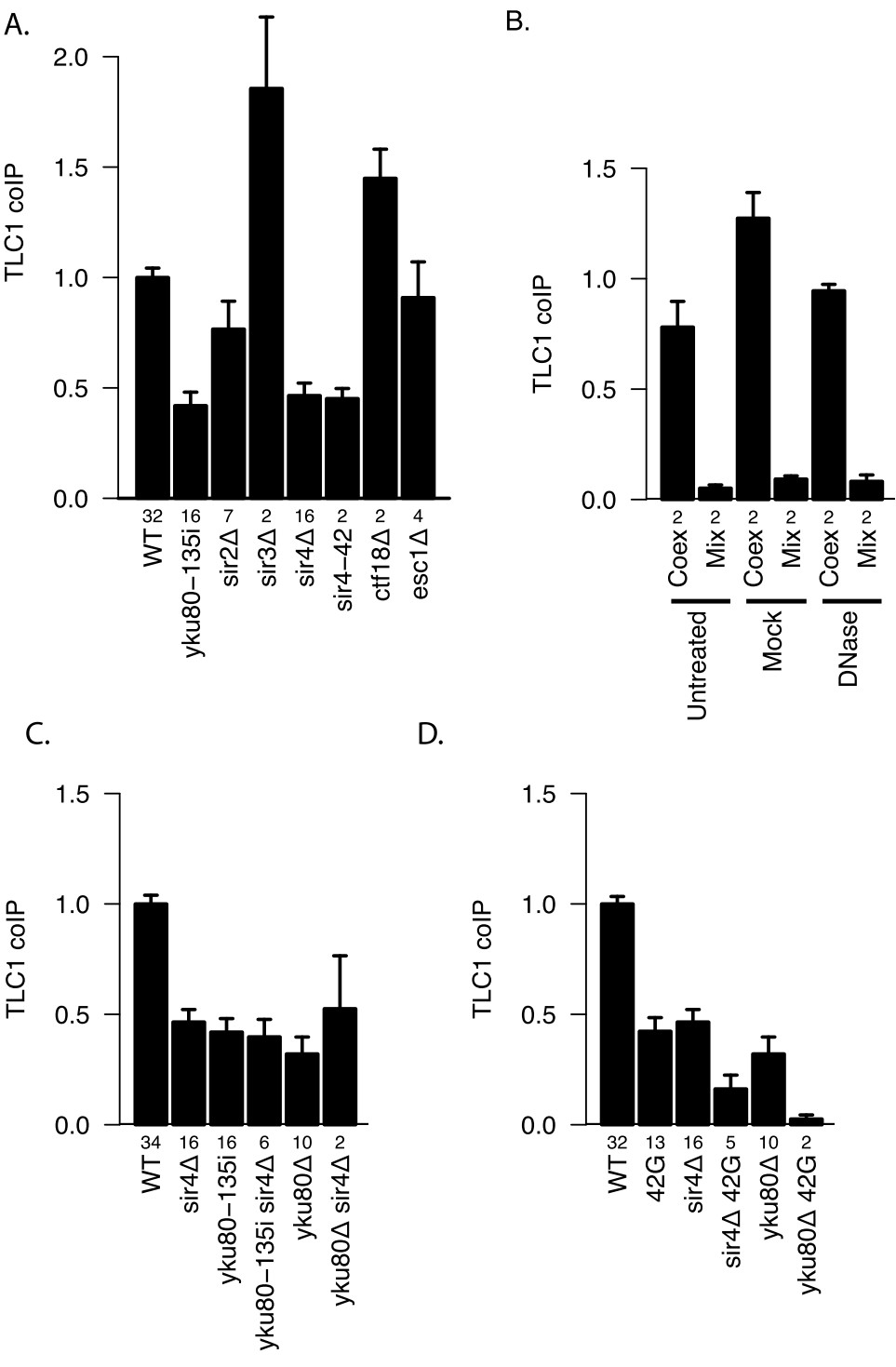

**Figure 6 Two separate pathways of TLC1-TLC1 association.** (A) Ku complex binding to TLC1 and Sir4 are required for TLC1-TLC1 association but telomere tethering to the nuclear periphery and telomere silencing are not. Mutations defective in either telomere tethering to nuclear periphery (*ctf18Δ* = yEHB22,764/765, and *esc1D* = yEHB22,766/767) or telomere (continued on next page...)

**Figure 6 (...continued)**
silencing ($sir2\Delta$ = yEHB22,728/729, yEHB22,819/820, $sir3\Delta$ = yEHB22,762/763, $sir4\Delta$ = yEHB22,730/731, yEHB22,817/818, and $sir4$-42 = yEHB22,787/788) are indicated. The fractions of TLC1 in dimer form, normalized to WT (yEHB22,662/663, yEHB22,807/808), are shown for strains indicated. The error bars indicate the standard error, and the numbers of experiments represented are shown just below the $x$-axis. (B) Lysate was either untreated, mock-treated, or treated with DNase prior to immunoprecipitation. The efficient loss of DNA was validated only in DNase treated samples. Despite the loss of DNA in the samples, the TLC1-TLC1 coIP efficiency was not reduced. WT strains (yEHB22,662/663) were used for "Coex" samples. In "Mix" samples, strains expressing tagged and untagged TLC1 independently (yEHB22,720/721 and yEHB22,722/723) were mixed before the lysis step. The fraction of TLC1 in dimer form, normalized to the average of "Coex" samples are shown. The error bars indicate the standard error, and the numbers of experiments represented are shown just below the $x$-axis. (C) The Ku mutations were combined with $SIR4$ deletion. The values are normalized to the average of the wild-type samples in each experiment. The error bars indicate the standard deviation among the samples. The fractions of TLC1 in dimer form, normalized to WT, are shown for strains indicated. WT = yEHB22,662/663, yEHB22,807/808, $sir4\Delta$ = yEHB22,730/731, yEHB22,817/818, $yku80$-135i = yEHB22,750/751, yEHB22,815/816, $yku80$-135i $sir4\Delta$ = yEHB22,821/822, $yku80D$ = yEHB22,662/663, $yku80\Delta sir4\Delta$ = yEHB22,774/775. The error bars indicate the standard error, and the numbers of experiments represented are shown just below the $x$-axis. (D) The Ku and Sir4 combined with the mutation in the 3' region. The deletion strains for $SIR4$ and $YKU80$ were combined with $tlc1$-42G (42G) mutations. The fractions of TLC1 in dimer form, normalized to WT, are shown for strains indicated. WT = yEHB22,662/663, yEHB22,807/808, 42G = yEHB22,742/743, yEHB22,809/180, $sir4\Delta$ = yEHB22,730/731, yEHB22,817/818, sir4D 42G = yEHB22,823/284, $yku80\Delta$ = yEHB22,662/663, $yku80\Delta$ 42G = yEHB22,776/777. The error bars indicate the standard error, and the numbers of experiments represented are shown just below the $x$-axis.

To test if the Ku complex and Sir4 act in the same pathway for TLC1-TLC1 association, we combined $sir4\Delta$ with $yku80\Delta$ or $yku80$-135i mutations. The double mutants showed no further reduction in the TLC1 dimer fraction compared to single mutants (Fig. 6C). We conclude that Ku binding to TLC1 and Sir4 regulates TLC1-TLC1 association through the same pathway, which is independent of telomere silencing or anchoring to the nuclear periphery.

### Ku/Sir4 and the 3'-cleaved TLC1 precursor sequence promote TLC1-TLC1 association by different modes

To determine the relationship between the roles of Ku/Sir4 and the 3' region of TLC1 in TLC1-TLC1 association, we combined $sir4\Delta$ or $yku80\Delta$ mutation with the 3' mutation $tlc1$-42G. In these double mutants ($sir4\Delta$ $tlc1$-42G and $yku80\Delta$ $tlc1$-42G strains), compared to either each single-mutant strain or the $sir4\Delta$ $yku80\Delta$ double mutant, the TLC1-TLC1 association was further reduced, down to almost to the background control level (Fig. 6D). This indicated that the TLC1-TLC1 association that is dependent on the 3' region of TLC1 is at least partially independent of Ku and Sir4, possibly mediated by a different pathway.

### Lack of evidence for Est2-Est2 physical association

Although, as described above, we did not find evidence that TLC1-TLC1 association was highly dependent on Est2, we tested the possibility that any of the small fraction of TLC1-TLC1 association that may be potentially affected by Est2 deletion might be mediated through association of one Est2 molecule with another Est2 molecule. To this end, we performed four different assays in attempts to detect any such physical Est2-Est2 interaction *in vivo*. First, we attempted to detect Est2-Est2 interaction by yeast two-hybrid

assay in which Est2 was fused to the Gal4 activation domain and DNA binding domain separately; such assays showed no positive signals for Est2-Est2 interaction. Secondly, we co-expressed Est2-FLAG and Est2-myc and performed co-immunoprecipitation assays; however, no signal indicative of co-immunoprecipitation was detected in the Western blots in these experiments. Thirdly, to overcome the potential issues of the detection limit using Western blotting, we performed coIP experiments using presence of TLC1 as a proxy signal, via qRT-PCR assays as described above. In this approach, we co-expressed wild-type Est2-HA with either wild-type Est2-myc (positive control) or *est2ΔCP*-myc. *est2ΔCP* deletes the Est2 domain that is required for Est2-TLC1 interaction (*Lin & Blackburn, 2004*). Therefore, the presence of an interaction between Est2-HA and *Est2ΔCP*-myc can be ascertained by proxy using the measurement of TLC1 in *est2ΔCP*-myc IP. However, we did not observe any such enrichment of TLC1 in this coIP assay (Fig. 7A). Finally, because TLC1 detection by the qRT-PCR assay had high sensitivity, we also performed sequential coIP experiments with strains co-expressing Est2-FLAG and Est2-myc. In this assay, Est2-FLAG was adsorbed onto anti-FLAG gel matrix and subsequently eluted with FLAG peptide, and any Est2-myc present in the elution fraction was immunoprecipitated with anti-myc antibody. The amount of TLC1 was then quantified in this final immunoprecipitate; while the positive control (Est2-FLAG-myc) showed robust enrichment, we found no enrichment of TLC1 compared to the negative control (Fig. 7B). We conclude that, although the possibility of a weak or transient association between Est2 molecules cannot be ruled out, these negative lines of evidence are consistent with the model that the majority of the TLC1-TLC1 *in vivo* association is independent of an active telomerase enzyme complex.

## DISCUSSION

Here we have explored the nature of telomerase RNA-RNA associations *in vivo* in *S. cerevisiae*. We report that ∼10% of the TLC1 molecules *in vivo* are physically associated with another TLC1 molecule. We refer to this as TLC1-TLC1 association for simplicity, although the data do not formally exclude the possibility of higher oligomerization forms. The lack of formation of TLC1-TLC1 association in lysates (the mix experiments) suggest that either TLC1 level is too low in the lysate or there is an active mechanism required for the association. This TLC1-TLC1 association increases by two-fold specifically in G1 phase of the cell cycle, and takes place via two distinguishable modes.

First, mutating a sequence in the 3′ region of TLC1 that is cleaved off during the production of the mature form of TLC1 reduced TLC1-TLC1 association by about half. The TLC1-TLC1 association of both the mature and the immature TLC1 forms were comparably affected by this 3′ sequence mutation. This same sequence has previously been implicated in TLC1-TLC1 association *in vitro* and its mutation shown to shorten telomeres (*Gipson et al., 2007*). Our findings thus indicate this 3′ sequence-dependent mode of TLC1-TLC1 association occurs *in vivo* during telomerase biogenesis. This is further consistent with our findings that TLC1-TLC1 association depends on nuclear export to the cytoplasm, where biogenesis of telomerase is reported to occur, and that TLC1-TLC1 association increases in G1 phase, the only time in the cell cycle when TLC1 maturation cleavage is active (*Chapon, Cech & Zaug, 1997*).

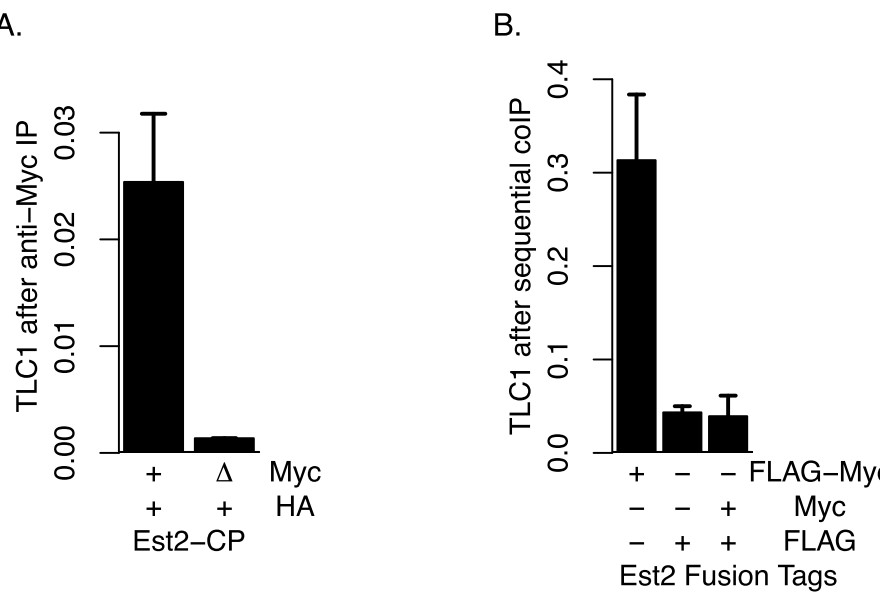

**Figure 7** **Lack of Evidence for Est2-Est2 association *in vivo*.** (A) Est2-ΔCP lacks the Est2 TLC1 binding domain. Myc-tagged Est2 (either Est2-Myc or Est2-ΔCP) was immunoprecipitated from lysates of strains co-expressing Est2-HA. Fractions of TLC1 isolated are shown. Error bars represent standard deviation from two experiments. The table below identify whether CP region of Est2 is deleted (Δ) in Myc or HA-tagged Est2. The strains used in these experiments were generous gift of Jue Lin (*Lin & Blackburn, 2004*). (B) The amount of TLC1 immunoprecipitated after sequential immunoprecipitation, anti-FLAG then anti-MYC, was measured. Amount of TLC1 remained in the MYC IP is represented as the fraction of TLC1 immunoprecipitated in the FLAG IP. The table below indicates EST2 fusions with specified tags present in each IP. The strains yEHB22,825-827 were used in these experiments.

The second mode of TLC1-TLC1 association requires Ku binding to TLC1; mutations preventing Ku-TLC1 interaction reduced TLC1-TLC1 association by about half. The Ku-associated protein Sir4 was also required for this mode. The Sir and Ku complexes are both important factors in maintaining telomeres; their functions include forming silent chromatin at telomeres and recruiting telomeres to nuclear periphery (*Boulton & Jackson, 1998*; *Taddei et al., 2004*). Interestingly however, although Sir4 is part of the silent information regulator Sir complex, TLC1-TLC1 association required neither classic silencing (neither Sir2 nor Sir3 was required), nor Ku-mediated telomere tethering to the nuclear periphery (neither Esc1 nor Ctf18 was required).

The additive genetic disruptions of these two modes of *in vivo* TLC1-TLC1 association - RNA sequence mutations in the 3′ region of TLC1 and deletion of the protein factors Ku and Sir4 - have an intriguing parallel to the *in vitro* disruptions of TLC1-TLC1 association in the immunoprecipitate, via either competition with excess oligonucleotides (most sensitive in the 3′ region) or protease treatment. Each of these two *in vitro* treatments disrupted only a fraction of the TLC1-TLC1 association. Combining these findings, the simplest interpretation is that these two fractions correspond to or overlap with the TLC1 3′ sequence-dependent and the Ku/Sir4 dependent association modes respectively.

Simultaneously mutating both the 3′ precursor TLC1 sequence and abrogating Ku-TLC1 binding abolished *in vivo* TLC1-TLC1 association to background levels. The epistasis

analyses together indicate that for physical TLC1-TLC1 association, Ku and Sir4 act in the same pathway, which is distinct from the pathway requiring the 3′ end sequence of the immature TLC1 RNA. Notably, each of the various kinds of mutations that we report here to impair TLC1-TLC1 association also causes telomeres to be shorter than wild-type (*Askree et al., 2004*), consistent with TLC1-TLC1 association *in vivo* having functional significance.

Our findings indicate two separable and potentially independent modes of TLC1-TLC1 association—the first involving the TLC1 3′ region prior to cleavage to the mature form, and a subsequent mode involving Ku/Sir4. We propose a model (Fig. 8) by which all TLC1 molecules transiently engage in TLC1-TLC1 association during at least two stages in telomerase biogenesis. The first TLC1-TLC1 association mode occurs prior to TLC1 maturation and requires a sequence in the 3′ extension of the TLC1 precursor (Fig. 8 Mode 1). The lack of compensatory mutation recue of TLC1-TLC1 association suggests the palindromic sequence in the 3′ region may be important for a recruitment of a factor or a secondary structure that is important in TLC1-TLC1 association rather than *trans* base-pairing. The TLC1-TLC1 association is further stabilized by RNA-RNA or RNA-protein interactions that persist after TLC1 cleavage/maturation, which can be partially disrupted *in vitro* by anti-sense oligonucleotides - particularly those complementary to the 3′ region of the mature telomerase RNA. Our findings suggest that multiple regions of TLC1 RNA help stabilize the TLC1-TLC1 association, and are consistent with a model of their "unzipping" caused by the addition of competing oligonucleotides.

The second mode requires Ku complex binding to TLC1 and also depends on Sir4 (Fig. 8 Mode II). While it is not known when in the biogenesis and maturation of TLC1 Ku (and possibly Ku-bound Sir4) become associated with TLC1, Ku and Sir4 are both thought to function at telomeres, where the vast majority of TLC1 (>95%) is already processed to the mature form (i.e., missing the 3′ region). Both mature TLC1 and uncleaved precursor TLC1 were found coIP'ed with Est2, albeit with the IP efficiency of the immature form being reduced by about half (Fig. 2G). Thus, cleaving off the 3′ region of TLC1 is not an obligatory step for TLC1 in order for it to engage in telomerase enzyme complex formation. This is consistent with the lack of interdependence we found between the 3′ sequence-mediated association during TLC1 biogenesis and the Ku/Sir-dependent association.

Interestingly, some of the data presented here cannot easily be reconciled with the data previously reported. Specifically, *Gipson et al. (2007)* reported that the compensatory mutations in the 3′ palindromic sequence (*tlc1-42G* and *tlc1-42C*) dimerized *in vitro* and rescued telomere shortening in *trans*; however, we observed that the *in vivo* TLC1-TLC1 association is not rescued by *trans* compensatory mutations. We observed no *in vitro* TLC1-TLC1 association in lysates (Fig. 2E), while Gipson et al. showed that high concentrations of *in vitro* transcribed TLC1 readily dimerized *in vitro*. These contrasting observations suggest that *in vitro* and *in vivo* TLC1-TLC1 associations may results from different mechanisms. The rescue of telomere length in strains co-expressing both *tlc1-42G* and *tlc1-42C* observed by Gipson et al. may have resulted from increased total TLC1 levels compared to strains expressing *tlc1-42G* or *tlc1-42C* alone. Hence, these observations are not directly contradictory; however further studies delineating the importance of *in*

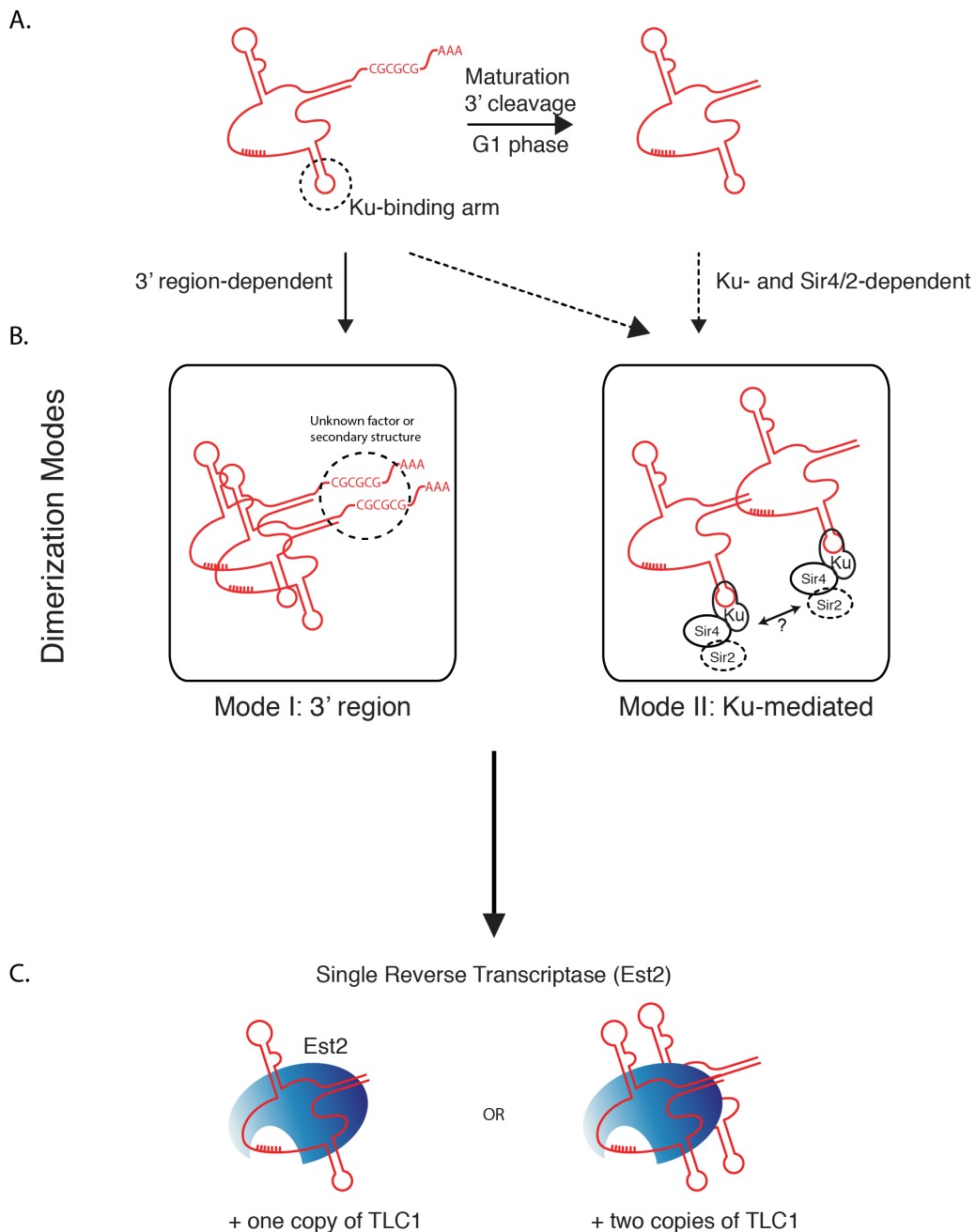

**Figure 8 Two modes of dimerization model.** (A) Schematic of TLC1 cleavage of 3′ region. Tick marks: template region of TLC1. CGCGCG: sequence at the 3′ region important in TLC1 dimerization. The stem-loop structure that the Ku complex binds is indicated. (B) Two modes of TLC1-TLC1 association *in vivo*. Mode I, dependent on the precursor TLC1 3′ region, is initiated before the 3′ region is cleaved off. Mode II, dependent on Sir4/2 and the Ku complex, possibly at telomeres. (C) TLC1 in the telomerase RNP is either monomeric or dimeric, but each RNP contains only one Est2.

*vivo* TLC1-TLC1 association with telomerase functions should clarify these seemingly contradictory findings.

Lin and Blackburn reported physical interactions between Est2 molecules by differentially tagging two copies of Est2 in coIP assays. The same strains were used in this study to test for presence of Est2-Est2 interaction by measuring TLC1 levels as a proxy. Surprisingly, in contrast to published results, TLC1 did not coimmunoprecipitate. It is possible that TLC1 only interacts with monomeric Est2, and that dimeric Est2's are inactive.

Finally, the presence of two independent modes and machineries for TLC1-TLC1 association suggest that such interaction reflects an important aspect of yeast telomere maintenance biology; a conclusion reinforced by the telomere shortening that results from all the mutations that disrupted TLC1-TLC1 association. However, this report leaves open the detailed mechanisms of these novel *in vivo* TLC1-TLC1 physical association modes that we have demonstrated in this study. One speculation is that these RNA-RNA associations may be important for the stability of telomerase RNA as it is shuttled among cytoplasmic and nuclear compartments for various maturation steps; a possible model is that TLC1-TLC1 association assists the RNA in acting as its own chaperone. We can further speculate that this might be an important regulatory step for telomerase activity, as the yeast telomerase holoenzyme shows no physical evidence of oligomerization. For example, a dissociation of TLC1-TLC1 association, which likely requires energy, may act as a switch mechanism for forming a fully competent telomerase holoenzyme. There are also other cell cycle regulated telomerase activation factors such as Est1 and Ku that recruit telomerase complex to the telomere at different stages of cell cycle, and it is of great interest to test how these factors may affect TLC1-TLC1 associations in cell cycle-dependent manner. Further research will be needed to decipher the mechanistic and functional significance of intermolecular interactions among telomerase components.

## ACKNOWLEDGEMENTS

The authors thank Tracy Chow, Beth Cimini, Kyle Jay, Jue Lin, Imke Listerman, and Dana Smith for critical reading of the manuscript and helpful discussion.

### Funding

This work was supported by an NIH grant (R01GM026259). The funders had no role in study design, data collection and analysis, decision to publish, or preparation of the manuscript.

### Grant Disclosures

The following grant information was disclosed by the authors:
NIH: R01GM026259.

### Competing Interests

The authors declare there are no competing interests.

## Author Contributions

- Tet Matsuguchi conceived and designed the experiments, performed the experiments, analyzed the data, wrote the paper, prepared figures and/or tables, reviewed drafts of the paper.
- Elizabeth Blackburn conceived and designed the experiments, contributed reagents/materials/analysis tools, wrote the paper, reviewed drafts of the paper.

## Data Availability

The raw data can be found in the Supplemental Information.

## Supplemental Information

Supplemental information for this article can be found online at http://dx.doi.org/10.7717/peerj.1534#supplemental-information.

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
