# Peer review of "The yeast telomerase RNA, TLC1, participates in two distinct modes of TLC1-TLC1 association processes in vivo"

_PeerJ, doi:10.7717/peerj.1534_

## Round 0.1 · original submission · Major Revisions

We have now received 2 reviews for your manuscript. Please address the critical points raised by both reviewers in your revision / rebuttal.

Reviewer 1 ·

Basic reporting

In the manuscript “The yeast telomerase RNA, TLC1, participates in two distinct modesl of TLC1-TLC1 association processes in vivo,” the authors address an ongoing debate in the telomerase field about the association of components of the telomerase complex (in this case, using yeast as a model). Ly and colleagues previously demonstrated that TLC1 RNA can dimerize in vitro. Here, Blackburn and Matsuguchi use a tagged version of the TLC1 RNA to conduct co-immunoprecipitation experiments. The information presented in this manuscript has the potential to contribute to the field, but the manuscript in its current version contains errors, inconsistencies and ambiguities that make it difficult to assess the results. I have included my explanation of these issues in the "Experimental design" section of the review.

Experimental design

1. I was left confused about which strains are utilized in each experiment. Despite an extensive strain table, there is no cross-referencing within the figure legends. For example, there are strains listed that contain two copies of the TLC1 gene (either both untagged, both tagged, or one of each). But there are also strains listed that apparently have only one copy of either tagged or untagged TLC1. When the strain is listed as “WT” in a figure, I’m not sure which strain it is (true WT, or two copies of the untagged TLC1)? What is the difference between yEHB22,321 and yEHB22,465? These two strains seem to have been carried through all manipulations, suggesting that they are two different isolates of the same strain. However, there is no mention of this detail in the Materials and Methods or how these strains were utilized in the experiments described here. What are the strains derived from yEHB22,804 and 805? When were they used in the manuscript? Rif1 and Rif2 deletion strains are listed in the table, but never used in the manuscript.

2. The text states in several places that the calculation for the fraction of TLC1 dimer is in the Materials and Methods. I can’t find it there. There is also no description of the cell synchronization methodology used for Figure 3.

3. I’m confused about how the data are produced and presented for the oligonucleotide competition experiments in Figure 2A and B. In the Materials and Methods, it is stated that the oligos were added to the wash buffer. Is that after the standard wash used in the other experiments? Was a standard IP done and a portion of that sample was analyzed and then the oligos were added for an additional wash period before removing a second sample? The legend to Figure 2A says that the “amount of TLC1 that remained in dimer is shown.” An “amount” of 1.0 doesn’t make sense.

Along similar lines, I’m confused in Figure 2B why the scale at the bottom is labeled “TLC1 coIP compared to WT.” The legend states instead that the values shown are “the fraction of TLC1 that remained on the beads after the wash.” I assume that this is relative to the amount on the beads prior to the wash (see above)?? What does WT mean in this situation?

4. Although the authors do indicate in the figure legend if they are using standard error or standard deviation, the number of times each experiment was repeated is often not clear. There are also very few places in which p values are given in the text. These values should be included (not just error bars).

5. In several figures, the legend states that the samples were normalized to the WT, but the averages shown for WT are not 1.0 (see for example Figure 4B, C, D, E, 5A). What am I missing? In Figure 4A, the authors state that the values were normalized to the average of trypsin-treated samples, but this must be “untreated” instead.

6. Referring to a strain that expresses two mutations in trans by the name 42GC is very confusing. I had to go to the strain table to confirm that this strain expresses both the 42G and 42C alleles in trans.

7. In the methods, the authors state that “MS2 coat protein fused to 3 Myc epitope tags was expressed ether in a tel1D or in experimental strains containing both tagged and untagged TLC1.” Despite a huge strain table, there is no strain deleted for TLC1 listed and the authors never make it clear when this strain was used. Were the co-IPs sometimes done with the MS2 coat protein expressed in the strain with the tagged RNA and sometimes added after capture on the beads? That’s what the methods section seems to indicate, yet there is no place in the figure legends or text where the authors clarify whether they are switching back and forth between techniques.

8. In the Southern blot shown in Figure 1F, there are no molecular weight markers. According to the methods, I think that the bottom band is a “marker” that was added, but the size is never given. From these data, I cannot evaluate the extent of the telomere shortening that is observed.

9. In Figure 1D, the strains utilized for the experiments must be clarified. There should also be controls showing that the PCR is specific (i.e. a strain with no MS2 insert gives only background signal and vice versa).

10. In the legend to Figure 5B, the authors state that Figure 5C shows the efficient loss of DNA only in DNA (sic) treated samples. However, these data do not exist and Figure 5C is something else. Not clear how the data in 5B were normalized.

11. Figure 6B. I can’t tell what the scale is on the Y axis. Fraction of total TLC1 RNA?

Validity of the findings

1. The model is confusing because the RNAs are drawn as base pairing between the palindromic regions, but the data in the manuscript and the figure legend say that this is not to be implied. What do the authors envision the role of these sequences to be? In the last part of the model, why are Ku and Sir4 shown associating only with one of the RNAs?

2. Lin and Blackburn reported in two supplementary figures that differentially tagged versions of Est2 interact with each other (Myc and HA tags). How do the authors explain this previous published result in light of the data presented here?

3. The authors argue that the telomere shortening previously reported for the palindrome mutants tlc1-42C and tlc1-42G suggests that the dimerization of TLC1 has functional significance. However, Ly and colleagues also reported that the coexpression of the two mutants (predicted to restore the intermolecular interaction) rescues the telomere length defect. In the current study, the authors find no rescue of the observed TLC1 interaction defect when the mutants are co-expressed. How do the authors reconcile this apparent contradiction?

Reviewer 2 ·

Basic reporting

See below

Experimental design

See below

Validity of the findings

See below

Additional comments

The authors are addressing an interesting question within the yeast telomere field on whether the long non-coding RNA TLC1, which minimally serves as the RNA template for the specialized reverse transcriptase enzyme telomerase, can form a dimeric RNA species. Overall, this is an interesting and fairly well body of work. In considering the study for publication the editor should weigh the following points:

1. The Southern blot data showing Y’ telomere DNA length in Figure 1F requires size markers as it is currently not possible to determine how much of an impact the addition of the MS2 sites to TLC1 has on the telomere length. The authors’ statement “slightly shorter but stable telomere lengths” is arbitrary. The quantified impact on telomere length should be determined. In addition, all telomeres should be evaluated not just Y’ telomeres.

2. The nucleic acid competitions shown in Figure 2 are difficult to rationalize given the inability of the full length TLC1 to “mix” with their TLC1-MS2 as shown in Figure 1G. If the full length TLC1 does not form an association then how do the oligonucleotides compete? Even if we assume the oligonucleotides are in higher excess, the full length TLC1 should eventually associate too. Do the authors detect mixing when the species are incubated for a longer time period?

3. The authors demonstrate that Est1 is important for the TLC1-TLC1 interaction. However, it appears the shown experiment was done using asynchronous cells. Is the impact of Est1 more significant if cell are sampled in S phase? Additionally, is the impact of Est1 limited to S phase or is it apparent in other phase of the cell cycle?

4. Are the Ku effects additive with the Est1 impact on the TLC1-TLC1 interaction?

---

## Round 0.2 · accepted · Accept

Thank you for addressing critical points of both reviewers and for revising the manuscript.